

# Brief communication: Comparison of thermistor and digital temperature sensor performance in a mountain permafrost borehole

Lars Widmer[1], Marcia Phillips[1], Chasper Buchli[1]

[1]WSL Institute for Snow and Avalanche Research SLF, Davos, Switzerland

*Correspondence to*: Lars Widmer (lars.widmer@slf.ch)

## Abstract

Monitoring mountain permafrost temperatures in boreholes is challenging regarding the resilience and long-term temperature stability of the sensor systems. Whilst resistance thermistors boast a high accuracy, they are prone to drift when exposed to moisture, pressure, or cable strain. Supplementing or replacing them with digital bandgap temperature sensors requires careful analysis of the sensor performance. We carry out a first comparison of two temperature sensor systems under field conditions in mountain permafrost, at 15 identical depths in one borehole. Temperature values, sensing delays and noise levels are compared and discussed.

## 1 Introduction

In mountain permafrost temperatures  need to be measured over several decades before climate effects can be quantified. High quality, accurate borehole temperature measurements are challenging due to slope deformation (Noetzli and Pellet, 2022), rising water contents (Phillips et al., 2020), rockfall and lightning (Noetzli et al., 2021). Since the late 1980s, resistance thermistors have been used in permafrost boreholes (mainly due to their high accuracy) for long-term monitoring (Permos, 2019; Haberkorn et al., 2021; Harris et al., 2001). However, thermistors are liable to drift (causing them to register erroneously high temperatures) if moistened or when the wires are stretched (causing erroneously low temperatures) (Luethi and Phillips, 2016). If initially water-tight tubes and cables are damaged with time, moisture penetrates the sensor system. Wire stretching or compression can be induced by borehole deformation. Ideally, sensor recalibration is carried out at regular intervals to determine whether drift is occurring. However, the extraction and recalibration of borehole sensors is often impossible due to borehole deformation. In this case, the performance of sensors located in the active layer can  be evaluated during the spring and/or autumn zero curtain, when ground temperature remains at 0°C during phase change. Due to the lack of a zero curtain beneath the active layer it is difficult to discern between the initial stages of drift and naturally induced warming/cooling (Luethi and Phillips, 2016) without recalibration or duplicate measurements.

Sensor duplication is thus ideal to ensure the long-term continuity and quality of ground temperature data (Noetzli et al., 2021), either using two identical systems but preferably with two separate and different ones, reducing the likelihood of simultaneous





failure. Ideally, one system consists of the hitherto well-tested resistance thermistors. Other types of temperature sensors need to be evaluated for their long-term suitability in this challenging environment.

Here we analyse the performance of two temperature sensing systems – thermistors and digital bandgap temperature sensors – which were installed in a tubed borehole in 2019 in a steep, creeping mountain permafrost talus slope. Our aim is to identify and quantify similarities/differences between the two systems and to determine their suitability for permafrost conditions.

## 1.1 Site description

Muot da Barba Peider is a NW oriented 38° talus slope at 2980 m a.s.l above Pontresina in the Eastern Swiss Alps. The rock consists of gneiss and the coarse-grained talus layer is around 1.5 to 2.0 m thick, fining downwards, with a volumetric ice content of around 10%. Active layer thickness varies across the slope and in recent years has been between 1.8 and 4 m thick (www.permos.ch). The underlying bedrock is permanently frozen and contains a little ice in cracks and pores. The talus creeps
around 10–20 cm per year (Phillips and Kenner, 2021).

## 1.2 Instrumentation and Methods

In 1996 two 20 m vertical boreholes (B1/MBP_0196 and B2/MBP_0296) were drilled and equipped with thermistors. Their data are available in the PERMOS database (www.permos.ch). As some of the thermistors in B1 and B2 were damaged or started to drift over time (Permos, 2019), a 20 m vertical borehole B3 (MBP_03_19 was drilled between B1 and B2 in 2019.
The aim was to secure the existing Muot da Barba Peider temperature data series with an initial overlap to allow for comparison. B3 was equipped with two types of temperature sensors, as specified in Table 1, installed in parallel in view of acquiring data for as long as possible. The borehole was furnished with a water-tight PVC tube. We installed a concrete chamber with an iron lid at the ground surface to house the data logger and the batteries. Sensors are at 0.5, 1.0, 1.5, 2.0, 2.5, 3.0, 3.5, 4.0, 5.0, 6.0, 8.0, 10.0, 13.5, 17.5 and 20.0-meters depth for both sensor strings. They were calibrated simultaneously
at 0°C in a double-walled ice-water bath, using the factory calibrated Steinhart-Hart equation for the thermistors and additional offset values for both sensor types.

The temperatures are logged using a Campbell CR1000X logger powered by two 12V, 12Ah lead-acid batteries. Measurements are taken at 30-minute intervals, averaged and saved in 2-hour intervals.

The data presented was obtained from 16 January 2020 to 22 August 2022. Unfortunately, gaps exist for both time series,
caused by battery or logger failure (Fig. 1d) and no full year of gapless data with both systems could be acquired yet. As both systems worked simultaneously during all seasons we can compare their performance during freezing, thawing, zero curtains and in periods with positive temperatures in the active layer.



**Table 1: Overview of differences and similarities between the two sensor types.**

| | Digital sensors | Analog thermistors |
|---|---|---|
| Type | Campbell CS225 [a] | Waljag Analog thermistor string [b] |
| Price | 5-10k CHF | 5-10k CHF |
| Availability | Custom order and production | Custom order and production |
| Sensor type | Band gap temperature reference with built-in 13-bit ADC, type ADT7410[c] | Epoxy encapsulated pressed ceramic disk NTC thermistor, type 44031RC[d] |
| Sensor casing | Rubber overmolding | Brass casing and heat shrink tubing |
| Metal cross-sectional area in wire and shielding | 1.31 mm² copper + 1.98 mm² T304 steel [e] | 0.5 mm² copper per sensor + 4.3 mm² copper shielding |
| Approximate self-heating power, per sensor and measurement | 12 mW during 10s [e] | 5 µW during 0.04 s |
| Minimum measuring time | 10s (Specification [a]) | 0.04s (Correspondence [f]) |
| Long term total system accuracy | ± 0.4°C (Specification [a]) | ± 0.1°C (Luethi and Phillips, 2016) |
| Resolution | 13-bit ADC, 0.0078 °C | Dependent on Logger |
| Effect of slight cable stretching | No effect expected due to digital transmission (Sdi-12, 2021) | Negative drift of sensors affected by stretching (Luethi and Phillips, 2016) |
| Effect of cable shearing | Loss of sensors below shearline, potential loss of all sensors due to short-circuit | Loss of sensors below shearline |
| Effect of moisture | Effect to be determined in further experiments / long-term use | Positive drift of affected sensor (Luethi and Phillips, 2016) |
| Effect of logger box temperature | No effect | Increased noise during summer |

[a] https://s.campbellsci.com/documents/us/manuals/cs225.pdf
[b] https://www.waljag.ch/angebot/permafrost/
[c] https://www.analog.com/en/products/adt7410.html#product-documentation
[d] https://www.te.com/usa-en/product-11026199-00.html
[e] Correspondence with Campbell Scientific
[f] Correspondence with Waljag GmbH





**Figure 1: Comparison of thermistor and digital sensor temperatures for different depths, above (a) and below 2.5 m depth (b).**

**c) Differences between the thermistor temperature data and the digital sensor data, over depth and time.**

**d) Valid temperature recordings available for the digital sensors and thermistors over time.**



## 2 Comparison of Sensor Performance

The direct comparisons (Figs. 1a, 1b) show good agreement between the sensor types with a correlation coefficient of 0.966. The dashed lines represent a difference of ±0.4°C between the sensors, which is the specified accuracy of the digital sensors. The data acquired in the active layer reveals that the digital sensors register approximately 5% lower temperatures above 0°C and 5% higher ones below compared to the thermistors. The agreement between the data values generally improves with depth (Fig. 1c). However, there are also some discrepancies between 10 and 17.5 m depth, particularly from January 2022 to August

2022, which remain to be explained once more data is available.

### 2.1 Comparison of Measured System Delay

To estimate the reaction delay between the two sensor types, the timeseries were filtered using a convolution with a Hann window (Harris, 1978) of width 24 values. Using a 2 h recording interval we obtained a 48 h smoothing window (see e.g. Fig. 2b). The digital sensors' timeseries was then refined by piecewise linear interpolation between all data points to obtain data

points at 0.1-hour intervals instead of 2 hours. The Pearson correlation coefficient was calculated using time-shifted versions of the digital sensors' and thermistors' timeseries. These are shown in Figure 2a for selected depths over a range of time shifts. Below 3.5 m temperatures change too slowly to determine the delay and are not shown. The maximum coefficient of correlation is achieved at 4.9, 3.2, 8.9 and 7.8 hours delay for the digital sensors at 0.5, 1.0, 1.5 and 2.0 m depth respectively.

### 2.2 Comparison of Measured System Noise

In Figure 2b the digital sensor signals look noisier than the thermistors' signals. The noise level was calculated for the entire measurement period as well as July to August 2020 (high logger box temperatures) and December 2021 to January 2022 (low logger box temperatures). The estimated standard deviation (SD) of both sensor types' noise was estimated using a method inspired by Sari et al. (2012).. The sensor time series is first filtered by convolving with the same 24-value wide Hann window as in section 2.1. The mean of the absolute difference between smoothed and measured values is multiplied by 1.253 to get the

estimated noise SD (Sari et al., 2012). The result is shown in Figure 2c. For a stationary signal, a long convolution window and purely gaussian noise, the procedure results in the exact noise SD. We assume these assumptions to be approximately met. Figure 2c shows that the near-surface sensors have higher estimated noise SD. This is likely an artefact, as they have a less stationary signal due to daily variations in temperature. In the overall noise SD estimate for sensors at 5 m depth or deeper, the digital sensors have an estimated noise SD of between $6 \times 10^{-3}$ °C and $4 \times 10^{-3}$ °C and the thermistors a noise SD between $2 \times$

$10^{-4}$ °C and $5 \times 10^{-5}$ °C. The logger box temperature is only relevant for the thermistors, as they display more noise in summer than in winter.







**Figure 2: a) Correlation coefficients of the thermistors and digital sensors vs applied time shift between the sensors. Maxima are marked with vertical ticks and can be read as the approximate delay of the digital sensor versus the thermistors.**

**b) Temperature measurements at 2.5 m depth between 1. Sept 2021 and 1. Nov 2021.**

**c) Estimated standard deviation SD of the estimated noise vs depth of the sensor. The digital sensors noise is clearly higher than the thermistors, but only the thermistors show a clear difference between summer and winter conditions.**



## 3 Discussion

The fact that both sensor systems shown here failed (either simultaneously or at different times) during the period between
2019 and 2022, as shown in Figure 1d underlines the importance of sensor duplication. The long-term performance as well as
the effect of moisture on the digital sensors is not yet known and will only be revealed in a few years.

For temperature monitoring in permafrost the accuracy of temperature sensors around 0°C is most important, so it is common
to rely on single point calibration at 0°C (Noetzli et al., 2021; Streletskiy et al., 2022). Multipoint calibration is significantly
more challenging, as it requires an alcohol bath and accurate reference thermometers instead of an ice-water bath. The
approximately 5% difference in slope between the thermistors and digital thermometers' temperatures found here underlines
that a multi-point calibration might improve the sensor performance and agreement significantly. White et al. (2014)
recommend a 2-point calibration for a temperature range of 10 °C with an expected uncertainty of about 0.1 °C. The digital
sensors register temperature changes with a delay on the order of 1-10 hours compared to the thermistors. This could have two
causes: a) higher insulation and thermal mass of the digital thermometers, and/or b) higher thermal conductivity along the
cable due to the larger copper cross-sectional area in the thermistors' cable. A delay can have disadvantages in ground layers
with daily temperature fluctuations – but the rapid reaction of the thermistors may not entirely reflect the true ground
temperature. During the summer months logger temperatures can produce noise in the thermistor data. Even then, the digital
sensors are significantly noisier than the thermistors, but the noise level is acceptable for both sensor types as it is significantly
lower than their respective accuracy. The measurement frequency could be increased to reduce noise, but has to be weighed
against the increase in self heating and power consumption.

## 4 Conclusions

We compare two temperature sensor systems (resistance thermistors and digital bandgap temperature sensors) in mountain
permafrost, at 15 identical depths in one borehole. Temperature values, sensing delays and noise levels are compared and
discussed. Overall, the differences between the temperature sensor types are minimal, with less than 5% of all values outside
of a 0.1°C difference. Since the risk of sensor failure in mountain permafrost boreholes is considerable, we can conclude that
it is best to measure with at least two temperature strings. Using different types of sensor strings further reduces the chance of
simultaneous failure.

We will collect and analyse a longer timeseries to characterize the measurement delay of the digital sensors more accurately.
Experiments should be carried out using identical encapsulation systems. Whereas the long-term behaviour of thermistors is
well known in mountain permafrost, we do not yet know how digital temperature sensors will react to moisture, cable stretching
or cable injury. To test the effects of calibration and linearity, both sensor types should be tested against a reference
thermometer in a laboratory setting. An experiment measuring heat conduction along the wires of the temperature strings
would be useful to quantify differences in reaction time. The self-heating of both sensor types should also be determined. This
first analysis of temperatures measured using two systems in a mountain permafrost borehole gives a first useful insight of
their advantages and disadvantages.



**Code and Data availability**

There is no code available and the data used is not publicly available.

**Contributions**

LW: Investigation, Methodology, Formal analysis, Visualization, Writing – original draft preparation

MP: Conceptualisation, Investigation, Methodology, Funding acquisition, Writing – original draft preparation

CB: Resources, Software, Writing – review & editing

**Competing interests**

The authors declare that they have no conflict of interest.

**Acknowledgements**

The authors thank the Swiss Permafrost Monitoring Network (PERMOS) and the Federal Office for the Environment (FOEN)
for funding for the drilling and instrumentation of the borehole. Canton Grisons is thanked for providing logistical support for
drilling. The companies Waljag and Campbell Scientific produced the temperature sensors and are thanked for their useful
advice during the design of the measurement setup. We are grateful to SLF electronics and mechanical workshop for their
technical support and Jeannette Nötzli for her thorough proofreading.

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
