# Peer review of "Brief communication: Comparison of thermistor and digital temperature sensor performance in a mountain permafrost borehole"

_EGUsphere, 2022_

## Author Response (AR1)

**Widmer et al. (EGUsphere 2022-1184) -** Brief communication: Comparison of thermistor and digital temperature sensor performance in a mountain permafrost borehole

**Author responses to the Reviewers**

Dear Reviewers,

Thank you very much for your constructive comments. Please find our responses and suggestions below (in blue).

With kind regards,

Lars Widmer, Marcia Phillips and Chasper Buchli

- **RC1**: 'Comment on egusphere-2022-1184', Philippe Schoeneich, 17 Mar 2023

The paper adresses a major problem of reliability and continuity of permafrost measurements in the context of long term monitoring. The paper is a technical paper, but is very clear for every one who is familiar with measureemnt techniques, who represent the targetted public.

One point would however need clarification: the stratigraphy f the borehole, section 1.1. Text mentions a talus layer around 1.5 to 2.0 m thick with a volumetric ice content of 10%, and an active layer of 1.8 to 4 m thickness over frozen bedrock. It is unclear if the talus is really only that thick (and thus bedrock from 1.5 m downwards), and if this is the case it is quite confusing how an active layer can have a volumetric ice content. An additional figure with the borehole section would be welcome, or at least a clarification of the text.

The borehole was drilled in autumn and moisture had already frozen back within the active layer. We will add '**approximately** 10%', as the volumetric ice volume in the active layer can vary from winter to winter. **(Changed in document)**

For the technical metrological part I have not the specific skills allowing a detailed comment.

**Citation**: https://doi.org/10.5194/egusphere-2022-1184-RC1

- **RC2**: 'Comment on egusphere-2022-1184', Ivar-Kristian Waarum, 19 Mar 2023

The brief communication paper presents a comparison of two different systems for measuring temperature in a borehole with permafrost conditions. Measurements from the two systems are compared with emphasis on difference in measured value, delay in measurement dynamics and measurement noise. The potential usefulness of the paper is undeniable, both for interpretation of temperature profiles in boreholes and for developing methodology for instrumentation of future boreholes.

Background, motivation and purpose of the paper is presented clearly.

The measurement frequency and parameters of the smoothing windows are stated. There is no mention of whether each measurement (every 30 minutes) is a single read from the ADC, or if each measurement is the result of an averaging over several reads. This type of averaging is fairly standard to implement to decrease electronic noise, but since noise is a focus in the paper it could be useful to state this for both measurement systems.

The temperature measurements are carried out every 30 minutes and the data averaged and stored every 2 hours. The borehole instrumentation was originally designed for long-term permafrost temperature monitoring (with two measurement systems to ensure long-term continuity). We later saw that a comparison of the data obtained using the two systems would be interesting. If we had planned this comparison from the start, we would have measured and stored the data every 30 minutes. **(No change)**

On line 69 you mention a discrepancy in the measurement values from Jan22 to Aug22. From Fig 1c/d it seems that the digital system did not record data between ca. Feb22 and Jul22. Do you mean that there is a discrepancy between data recorded prior to Feb22 and after Jul22? In any case, it would be interesting to read what the reason was for the gap in data from the digital system, since this could shed some light on operational differences between the two systems.

*We mention here that more data would make it easier to understand the greater discrepancies in 2022.* **We now have data until 12.03.2023 and would like to suggest showing it in the paper (see for example new Figure 1c below), as it is particularly interesting. Our interpretation***:*

The discrepancies between the two sensor systems are highest during the summer 2022 heat wave and diminish again afterwards. They are even high at 20 m depth, during a period when the thermal effects of the heat wave cannot yet have an effect at depth due to the thermal lag and delay with increasing depth.

As air and near-surface temperatures were much higher than in Summer 2022 than during the previous two summers, we surmise that the differences may be related to the influencing factors presented in the Discussion (e.g. logger temperature, thermal conductivity of measurement system). This is supported by the fact that the differences diminish again in autumn and winter 2022-2023.

[Figure]

Fig. 1c (NB: now with data until March 2023)

If you agree, we will add the new data to the paper and make the appropriate adjustments in the figures and text (dates etc.). There are no major differences in our findings other than the

observations mentioned above. **(Changed in document, chapter 2 as well as multiple places where the date was mentioned)**

On line 86 there is a mention of some 'assumptions that are assumed to be approximately met'. If these assumptions are important for the method applied here, it would be good to state more clearly how or to which degree they are met.

Agreed, we will change

"For a stationary signal, a long convolution window and purely gaussian noise, the procedure results in the exact noise SD. We assume these assumptions to be approximately met."

to

" For a stationary signal, a long convolution window and purely gaussian noise, the procedure results in the exact noise SD. The borehole temperature changes slowly and we assume the measurement noise to be of gaussian nature." **(Changed in chapter 2.2)**

I think the discussion section is very good, also that you mention possible reasons for the delay in the dynamics of the digital system. From Fig 2b it seems that the delay in measurement dynamics are only present when the temperature sinks. If this is the case in general, it would be interesting to read how it could relate to the mentioned reasons, as well as to the self-heating of sensors that you mention in the conclusion.

The situation shown in Fig. 2b is not always the case. Sometimes the delay occurs whilst temperatures are rising (e.g. at 2.5 m on 10/2020) **(no change in document)**

[Figure]

The figures are expressive and complements the text very well. The overall content and structure of the paper is concise and efficient.

Minor comments:

36 – The slope with the borehole is referred to with the name of the peak

We don''t understand the question/comment? **(no change in document)**

Fig 1d – Typo in heading

Will change to: Data **availability  (changed in document)**

Additional changes:

- Updated Correspondence E-mail address
- Code and Data availability updated
- Acknowledgements updated to include reviewers and editor.

---

## Editor Decision (ED1)

[revised manuscript text omitted]

---

## Author Response (AR2)

Hello!

Thanks for the detailed answer, here are our changes according to your comments (in blue).

Please include your response to the reviewer comment also here in the text. Just adding "approximately" is not enough of a clarification as requested by this reviewer.
Line 38: added "in autumn, when moisture in the active layer freezes."

I agree that you include your new data. However, I also would like you to answer (in the text) to the reviewers comment concerning the data gap of the digital system.
The gaps in the data (for both systems) are explained on Line 54. The discrepancies in the data from the two systems are addressed in Line 71 onwards and in the Discussion, on Line 110 onwards. We therefore think that we answered this question.

Please include relevant parts of your response to the reviewer here in the text, rather than just explaining it in your response to the reviewers
Line 81: added "We observe similar delays both when temperature rises and falls."

I suggest to rephrase: The thermistor data from the borehole is available through www.permos.ch (borehole MBP_0319). The data from the digital temperature sensors can be acquired from Marcia Phillips (phillips@slf.ch).
Agreed, we changed it according to your suggestion.